# Walking the Values in Bayesian Inverse Reinforcement Learning

**Ondrej Bajgar**[1]     **Alessandro Abate**[1]     **Konstantinos Gatsis**[2]     **Michael A. Osborne**[1]

[1]University of Oxford
[2]University of Southampton

## Abstract

The goal of Bayesian inverse reinforcement learning (IRL) is recovering a posterior distribution over reward functions using a set of demonstrations from an expert optimizing for a reward unknown to the learner. The resulting posterior over rewards can then be used to synthesize an apprentice policy that performs well on the same or a similar task. A key challenge in Bayesian IRL is bridging the computational gap between the hypothesis space of possible rewards and the likelihood, often defined in terms of Q values: vanilla Bayesian IRL needs to solve the costly forward planning problem – going from rewards to the Q values – at every step of the algorithm, which may need to be done thousands of times. We propose to solve this by a simple change: instead of focusing on primarily sampling in the space of rewards, we can focus on primarily working in the space of Q-values, since the computation required to go from Q-values to reward is radically cheaper. Furthermore, this reversion of the computation makes it easy to compute the gradient allowing efficient sampling using Hamiltonian Monte Carlo. We propose ValueWalk – a new Markov chain Monte Carlo method based on this insight – and illustrate its advantages on several tasks.

## 1 INTRODUCTION

Reinforcement learning (RL) has shown impressive performance across a wide variety of tasks, ranging from robotics to game playing. However, one of the main challenges in applying RL to real-world problems is specifying an appropriate reward function by hand, which is often difficult and can result in reward functions that are only imperfect proxies for designers' intentions. Inverse reinforcement learning (IRL) addresses this issue by instead learning the underlying reward function from expert demonstrations.

A key challenge in IRL is that the reward function is often underdetermined by the available demonstrations, as multiple reward functions can lead to the same optimal behaviour. This can be solved by picking a criterion for choosing among the reward functions compatible with the demonstrations – maximum margin [Ng and Russell, 2000, Ratliff et al., 2006] and maximum entropy [Ziebart et al., 2008] are the most prominent examples. As an alternative, Bayesian IRL explicitly tracks the uncertainty in the reward using a probability distribution. This not only accounts for the issue of underdeterminacy but also provides principled uncertainty estimates to any downstream tasks, which can be used, for instance, for the synthesis of safe policies or for active learning.

While having these attractive properties, Bayesian IRL is computationally challenging. While inference is done over the space of reward functions (in terms of which the prior is also expressed), the likelihood is usually formulated in terms of Q values (or is otherwise linked to the distribution of trajectories), and going from the former to the latter may require solving the whole forward planning problem at each iteration (as is case in the original Bayesian IRL algorithm [Ramachandran and Amir, 2007]), which is expensive in itself and may further need to be done thousands of times during IRL inference. To avoid this, we propose to use a simple insight: while going from rewards to Q-values is expensive, the inverse calculation can be much simpler. Thus, we propose to perform the inference as if it were done primarily over the space of Q-values, computing reward estimates beside it, resulting in a much cheaper algorithm. A related formulation appeared already in the variational method of Chan and van der Schaar [2021], which was, however, learning only a point estimate of the Q-function thus sacrificing Bayesianism from the centre of the algorithm.

We instead propose a new method that provides a full Bayesian treatment of the Q values, along with the rewards,

*Accepted for the 40th Conference on Uncertainty in Artificial Intelligence* (UAI 2024).

and is able to provide samples from the true posterior, being based on Markov chain Monte Carlo (MCMC) as opposed to variational inference, which needs to pre-specify a family of distributions within which to approximate the posterior. Furthermore, since the computation required at each step is much simpler than in prior MCMC-based methods [Ramachandran and Amir, 2007, Michini and How, 2012], which in itself makes our method more efficient, we can also easily calculate the gradient, which allows us to use Hamiltonian Monte Carlo [Duane et al., 1987] granting further gains in efficiency.

The contributions of this paper are the following: (1) we provide the first MCMC-based (and thus agnostic to the shape for the posterior) algorithm for continuous-space Bayesian inverse reinforcement learning; (2) we show that it scales better on discrete-space cases than the MCMC-based baseline, PolicyWalk; and (3) we show that we outperform the previous state-of-the-art algorithm for continuous state-spaces, AVRIL, better capturing the posterior over rewards and performing better on imitation learning tasks.

The paper is organized as follows: Section 2 provides background on inverse reinforcement learning and Hamiltonian Monte Carlo and summarizes related work. Section 3 introduces our proposed algorithm called ValueWalk. Section 4 compares our approach to an MCMC-based predecessor, PolicyWalk [Ramachandran and Amir, 2007], the previous state-of-the-art scalable method for Bayesian IRL, AVRIL [Chan and van der Schaar, 2021], and 2 imitation learning baselines on several control tasks.

## 2 BACKGROUND

### 2.1 BAYESIAN INVERSE REINFORCEMENT LEARNING

The goal of Bayesian inverse reinforcement learning is recovering a posterior distribution over reward functions based on observing a set of demonstrations $\mathcal{D} = \{(\phi(s_1), a_1), ..., (\phi(s_n), a_n)\}$ from an expert acting in a Markov decision process (MDP) $\mathcal{M} = (\mathcal{S}, \mathcal{A}, p, r, \gamma, t_{\max}, \rho_0)$ where $\mathcal{S}, \mathcal{A}$ are the state and action spaces respectively, $\phi : \mathcal{S} \to \Phi$ is a feature function representing states in a feature space $\Phi$, $p : \mathcal{S} \times \mathcal{A} \to \mathcal{P}(\mathcal{S})$ is the transition function where $\mathcal{P}(\mathcal{S})$ is a set of probability measures over $\mathcal{S}$, $r : \Phi \times \mathcal{A} \to \mathbb{R}$ is a reward function, $\gamma \in (0, 1)$ is a discount rate, $t_{\max} \in \mathbb{N} \cup \{\infty\}$ is the time horizon, and $\rho_0 \in \mathcal{P}(\mathcal{S})$ is the initial state distribution.

In IRL, we know all elements of the MDP except for the reward function and, possibly, the transition function (the setting without the knowledge of transition dynamics – or other form of access to the environment or its simulator – is sometimes called *strictly batch* [Jarrett et al., 2020]; our method is applicable in both this setting and the one

including an environment simulator, though most of the experiments are run in the former setting following the main baseline method, AVRIL). Instead, we have a model of how the expert policy is linked to the reward and, in the case of Bayesian IRL, also a prior distribution over reward functions, $p_R$ (which is, in general, a multi-dimensional stochastic process, that for any set of state-action pairs returns a joint probability distribution over the corresponding set of real-valued rewards). Commonly used expert models include Boltzmann rationality models such as

$$\mathbb{P}[a_i|\phi(s_t)] = \frac{e^{\alpha Q^*(\phi(s_t), a_i)}}{\sum_{a' \in \mathcal{A}} e^{\alpha Q^*(\phi(s_t), a')}} \quad (1)$$

[Ramachandran and Amir, 2007, Chan and van der Schaar, 2021] where $Q^*(s, a)$ is the expected (discounted) return if action $a$ is taken in state $s$, and the optimal policy is subsequently followed, and $\alpha$ is a rationality coefficient; the maximum entropy approach [Ziebart et al., 2008], where the probability of each trajectory is assumed to be proportional to the exponential of the trajectory's return; or sparse behaviour noise models [Zheng et al., 2014], where the expert is assumed to behave rationally except for sparse deviations. Beside these approximately rational models, various models of irrationality can also be considered [Evans et al., 2015]. The Bayesian IRL framework is flexible with respect to the choice of expert model, each such model just resulting in a different likelihood function, and can also be extended to the case where the model is not fully known.

In this article, we adopt the Boltzmann rationality model (1). We will assume that conditional on the Q values, the actions chosen by the expert are independent, yielding the likelihood

$$p(\mathcal{D}|r) = \prod_{s_t, a_t, s_{t+1} \in \mathcal{D}} \frac{e^{\alpha Q^*(\phi(s_t), a_t)}}{\sum_{a' \in \mathcal{A}} e^{\alpha Q^*(\phi(s_t), a')}} p(s_{t+1}|s_t, a_t) \quad (2)$$

for a discrete action space $\mathcal{A}$ (the expression can readily be adapted to a continuous setting by replacing the sum by an integral). Given this likelihood together with the prior over rewards $p_R$, we can calculate the posterior using the Bayes Theorem as $p(r|\mathcal{D}) = p(\mathcal{D}|r)p_R(r)/p(\mathcal{D})$. Generally, we cannot calculate this posterior analytically, so in practice, we need to resort to approximate methods. In this article, we use Markov chain Monte Carlo sampling.

When performing Bayesian inference over the reward, the transition probabilities will be considered fixed (except for Appendix A, which discusses the extension of Bayesian inference also to transition probabilities). Thus looking at the likelihood as a function of the reward, we can write

$$p(\mathcal{D}|r) = c \prod_{s_t, a_t \in \mathcal{D}} \frac{e^{\alpha Q^*(\phi(s_t), a_t)}}{\sum_{a' \in \mathcal{A}} e^{\alpha Q^*(\phi(s_t), a')}} =: c\mathcal{L}(\mathcal{D}|r). \quad (3)$$

Since $p(D) = \int p(D|r) dp_R(r) = c \int \mathcal{L}(D|r) dp_R(r)$, the constant transition term cancels out in the posterior, and,

going forward, we can use the partial likelihood $\mathcal{L}$ in reward posterior inference. Furthermore, MCMC algorithms generally depend only on the unnormalized distribution, thus we can also drop the remainder of the marginal $p(D)$ from our calculation.

## 2.2 MARKOV-CHAIN MONTE CARLO (MCMC)

Markov chain Monte Carlo (MCMC) methods form a class of algorithms widely used for sampling from complex probability distributions. MCMC methods rely on constructing Markov chains whose stationary distribution is the distribution of interest. Usually a new candidate sample in the chain is proposed and then accepted or rejected with probability proportional to the one under the target distribution – in our case the posterior over rewards.

In simpler MCMC methods, such as Metropolis-Hastings [Metropolis et al., 1953, Hastings, 1970], which were also used in some previous articles on Bayesian IRL [Ramachandran and Amir, 2007, Michini and How, 2012], the new step is proposed as a random jump in the sampling space. However, this often leads to a high rejection rate, if the jumps are large, or tightly correlated samples, if the jump is small, both of which can make the algorithm inefficient.

Thus, we instead use the popular Hamiltonian (or hybrid) Monte Carlo (HMC; Duane et al. [1987]) with the no-U-turn (NUTS) sampler [Hoffman and Gelman, 2014], which uses the gradient of the posterior density and Hamiltonian-like dynamics to propose samples that are far apart but still likely under the posterior, keeping a high acceptance rate, thus improving the efficiency of the algorithm.

## 2.3 RELATED WORK

Inverse reinforcement learning is most often used as a component in imitation learning: the more general task of learning an apprentice policy from expert demonstrations (see Zare et al. [2023] for a good recent survey). Beside IRL, the other major family of methods within imitation learning is behavioural cloning [Pomerleau, 1991, Ross et al., 2011], which, in its vanilla form, aims to learn the policy via supervised learning directly from the expert's observation-action pairs. The supervised learning approach has an advantage of lower computational cost, but faces the challenge of covariate-shift, since the training states are distributed according to the expert policy, not that of the learner agent, though multiple methods try to mitigate this by encouraging the learner policy to stay close to the expert one [Dadashi et al., 2020, Reddy et al., 2019, Brantley et al., 2019].

Inverse reinforcement learning represents an alternative which, instead of directly learning the observation-action mapping, first learns an estimate of the reward function, which can then be used to synthesize a policy. This can offer better generalization, but usually requires a model of the environment or access to it in order to run reinforcement learning, and generally incurs a higher computational cost.

We build on the paradigm of Bayesian IRL introduced by Ramachandran and Amir [2007]. While the Bayesian approach is attractive thanks to its principled treatment of uncertainty in light of the limited demonstration data, the key downside relative to other methods has been its scalability to higher-dimensional settings. Michini and How [2012] try to improve efficiency upon Ramachandran by focusing computation into regions of the state space close to the expert demonstration, still using MCMC, while Chan and van der Schaar [2021] try to improve efficiency by using an approximate variational distribution to model the posterior, as well as an additional neural network that tracks the Q function, which avoids the need for a costly inner-loop solver. Mandyam et al. [2023] has recently used kernel density estimation as an alternative method for approximate Bayesian inference.[1]

As opposed to recent work experimenting with other approximation techniques, we return to MCMC, with its greater expressivity, while at the same time adapting it to be used with continuous state spaces, which would not be feasible with prior MCMC-based methods.

## 3 METHOD

Similarly to early work in Bayesian IRL [Ramachandran and Amir, 2007, Michini and How, 2012], we use Markov chain Monte Carlo sampling to produce samples from the posterior distribution over rewards given a prior and expert demonstrations. Our key innovation is in the way we calculate the posterior. At each step of the Markov chain, these previous methods generally (1) proposed a new reward (2) used some method of forward planning, such as policy iteration, to deduce the corresponding optimal Q function and then (3) used the Q function to evaluate the likelihood and the reward to evaluate the prior.

We suggest proceeding the other way round: our method proposes a set of new parameters of the Q function and then uses it to deduce the corresponding rewards, which is generally a much easier calculation than going from rewards to Q functions. The method then uses the reward to calculate the prior and the Q value to evaluate the likelihood, and combines the two to calculate the unnormalized posterior density. This value can then be used for calculating the acceptance probability in an MCMC algorithm. Also thanks to the calculation being simple (rather than involving a RL-like inner-loop problem) and differentiable, we can also calcu-

---

[1]The evaluation in this paper focuses on an offline setting without access to environment dynamics, while the last mentioned method fundamentally depends on having access to the environment dynamics so we omit it from the comparison in this paper.

late the gradient, which we can use for efficient proposals using HMC+NUTS. Since we construct the random chain in the space of Q values instead of the space of rewards, used by previous methods, we call our new method ValueWalk.

## 3.1 FINITE STATE AND ACTION SPACES

Let us first outline the algorithm for the case of finite state and action spaces since the calculation can be performed exactly in this case, and the later continuous algorithm builds on this base case. We concentrate here on the calculation of the posterior probability corresponding to a single proposed set of Q values (which is performed at each step of the HMC trajectory) and otherwise employ standard HMC. Note that here, we assume the knowledge of the environment dynamics $P$, since this finite setting is close to that of PolicyWalk [Ramachandran and Amir, 2007], which also assumes this knowledge. However, the method can easily be extended to the *strictly batch* setting using steps analogous to the ones taken in the next subsection on continuous spaces.

In this finite case, we maintain a vector $Q \in \mathbb{R}^{|\mathcal{S}||\mathcal{A}|}$ representing the Q-value for each state-action pair. The first thing to notice is that given such a vector, we can calculate the corresponding reward vector of the same dimensionality as $Q$ using the Bellman equation as

$$R(s,a) = Q(s,a) - \gamma \sum_{s' \in \mathcal{S}} p(s'|s,a) \sum_{a' \in \mathcal{A}} \pi_Q(a'|s') Q(s',a')$$
(4)

with either $\pi^Q(a'|s') = \mathbb{I}[a' = \text{argmax}_{a''} Q(s',a'')]$ or a softmax approximation (which we use since it has the advantage of being differentiable using an inverse temperature coefficient $\bar{\alpha}$ to regulate the softness of the approximation). Equation (4) can also be written in vector form as $R = (I - \gamma \bar{P})Q$ where $\bar{P}$ is a $|\mathcal{S}||\mathcal{A}| \times |\mathcal{S}||\mathcal{A}|$ matrix whose values are defined as $\bar{P}(s,a;s',a') = P(s'|s,a)\pi^Q(a'|s')$. In that case, given a prior $p_R$ over rewards, we can calculate the prior of $Q$ as

$$p_Q(Q) = p_R((I - \gamma \bar{P})Q) \det(I - \gamma \bar{P}),$$

where $p_Q$ and $p_R$ are the prior probability densities of $Q$ and $R$ respectively. Since $\bar{P}$ is a stochastic matrix and $0 < \gamma < 1$, the determinant is always strictly positive.

This can be combined with the likelihood

$$\mathcal{L}(D|Q) = \prod_{(s,a) \in \mathcal{D}} \exp(\alpha Q(s,a)) / \sum_{a' \in \mathcal{A}} \exp(\alpha Q(s,a'))$$

to calculate the unnormalized posterior density $p(Q|\mathcal{D}) \propto p_Q(Q)\mathcal{L}(\mathcal{D}|Q)$ which we use in the standard HMC+NUTS algorithm to produce samples from the posterior. Note that the algorithm takes form of sampling Q-values, but produces samples of rewards as a byproduct, which is what we are

---

**Algorithm 1:** Calculation of the unnormalized posterior for finite $\mathcal{S}$ and $\mathcal{A}$ and known transition probabilities $P$ (performed in each step of HMC). The resulting candidate reward sample $\bar{R}$ is then accepted/rejected together with the corresponding Q.

**Data:** a candidate matrix of Q values, set of expert demonstrations $\mathcal{D}$, prior over rewards $p_R$

1 **for** $s, s' \in \mathcal{S}, a, a' \in \mathcal{A}$ **do**
2     $\pi^Q(a'|s') = \mathbb{I}[a' = \text{argmax}_{a''} Q(s',a'')]$ ;
3     $\bar{P}(s,a;s',a') = p(s'|s,a)\pi(a'|s')$ ;
4 **end**
5 $\bar{R} = (I - \gamma \bar{P})\bar{Q}$ where $\bar{R}, \bar{Q}$ are flattened vector versions of the reward and Q-value matrices ;
6 $p_Q(Q) = p_R(\bar{R}) \det(I - \gamma \bar{P})$ ;
7 $\mathcal{L}(\mathcal{D}|Q) = $ $\prod_{(s,a) \in \mathcal{D}} \exp(\alpha Q(s,a)) / \sum_{a' \in \mathcal{A}} \exp(\alpha Q(s,a'))$ ;
**Result:** $p(Q|\mathcal{D}) \propto p_Q(Q)\mathcal{L}(\mathcal{D}|Q)$; candidate sample $\bar{R}$

---

primarily interested in. Algorithm 1 summarizes this calculation. Note that Q here corresponds to the optimal Q value (as opposed to the one corresponding to the expert policy).

Theorem 1 in Appendix B formally proves that even though the algorithm primarily performs MCMC sampling over Q values, the secondary Markov chain over rewards produced by the algorithm also satisfies the detailed balance condition with respect to the posterior over rewards and thus constitutes a valid MCMC algorithm for sampling from the reward posterior.

Note that the determinant needs to be recalculated only if the optimal policy changes. Furthermore, we found that in practice, the recovered samples do not differ significantly if the determinant term is omitted.

See Section 4.1 for an example of this finite-case algorithm applied to a gridworld environment. Note that if the reward is known to depend only on the state, the sampling can instead be performed over state-values $V$. Similarly, if it depends on the full state, action, next state triple, it should be performed over state-action-state values to maintain a match in the dimensionality of the reward and value spaces.

The algorithm (and the Q-space trick) extends to the case of unknown transition probabilities. See Appendix A for more details on this.

## 3.2 CONTINUOUS STATE REPRESENTATIONS

For continuous or large discrete spaces, it is generally no longer possible or practical to maintain a separate Q-function parameter for each state, so we need to resort to approximation. Thus, from now on, our inference will centre around parameters $\theta_Q \in \mathbb{R}^{n_Q}$ of a Q function approximator

$Q_\theta : \Phi \times \mathcal{A} \to \mathbb{R}$ where $\Phi$ is the space of feature representations of the states. While the method is again centred around the Q function, the algorithm can also produce samples from the *reward* posterior at any set of evaluation points of interest, $\mathcal{D}_{\text{eval}}$. Furthermore, a method such as warped Gaussian processes [Snelson et al., 2003] can then be used to generalize the reward posterior from $\mathcal{D}_{\text{eval}}$ to new parts of the state-action space.

The likelihood calculation remains very similar to the discrete case:

$$\mathcal{L}(\mathcal{D}|\theta_Q) = \prod_{(s,a) \in \mathcal{D}} \frac{\exp(\alpha Q_{\theta_Q}(\phi(s), a))}{\sum_{a' \in \mathcal{A}} \exp(\alpha Q_{\theta_Q}(\phi(s), a'))} \quad (5)$$

(assuming $\mathcal{A}$ to be bounded). What concerns the evaluation of the prior, the reward corresponding to given Q-function parameters can be expressed using the continuous Bellman equation as

$$R(s,a) = Q_{\theta_Q}(\phi(s), a) - \gamma \mathbb{E}_{s',a'|s,a}\left[ Q_{\theta_Q}(\phi(s'), a') \right]$$

on any subset of states and actions.

In general, the integral in $\mathbb{E}_{s',a'|s,a}[Q_{\theta_Q}(\phi(s', a')] = \int_{s' \in \mathcal{S}} p(s'|s,a) \max_{a' \in \mathcal{A}} Q_{\theta_Q}(\phi(s'), a')$ needs to be approximated, for which any of a number of numerical methods can be used, from grid sampling to Monte Carlo methods, to more sophisticated techniques like probabilistic numerics [Hennig et al., 2022]. For most of these methods, we approximate the integral using a discrete set of candidate successor states $S_{\text{succ}}(s,a) = \{s \sim q(\cdot|s,a)\}$ sampled from some proposal distribution $q$ and then approximate the integral by

$$\frac{1}{|\mathcal{S}_{\text{succ}}|} \sum_{s' \in \mathcal{S}_{\text{succ}}} \frac{p(s'|s,a)}{q(s'|s,a)} \max_{a' \in \mathcal{A}} Q_{\theta_Q}(\phi(s'), a'). \quad (6)$$

The variant of the approximation we choose depends of what information we have at our disposal:

- If we have access to a probabilistic model $\hat{p}$ of the environment (which can either represent the true environment dynamics, if we know them, or our best inferred model of the dynamics including any epistemic uncertainty) that we can sample from, we can simply sample $\mathcal{S}_{\text{succ}}(s,a) = \{s' \sim \hat{p}(\cdot|s,a)\}$ and drop the importance weight.

- If we can evaluate the density $\hat{p}$ we can directly use the importance sampling equation 6 with $q$ being a proposal distribution ideally close to $\hat{p}$.

- If all we have is a static set of trajectories $\mathcal{D}_+$ – either just the expert ones $\mathcal{D}$, or also additional ones sampled from another, possibly random, policy – we can crudely approximate the reward for a transition $s, a, s' \in \mathcal{D}_+$ using a singleton $\mathcal{S}_{\text{succ}}(s,a) = \{s'\}$. This is an approximation made by the baseline AVRIL algorithm, so to

---

**Algorithm 2:** Calculation of the unnormalized posterior probability with continuous state representations for a single proposed parameter value $\theta_Q$ (performed in each step of MCMC). The returned candidate reward samples are accepted or rejected by the outer MCMC algorithm together with the candidate parameters $\theta_Q$.

**Data:** candidate parameters of the Q-function $\theta_Q$, a set of expert demonstarations $\mathcal{D}$, a set of evaluation locations $D_{\text{eval}}$, prior over rewards $p_R$

**1** Initialize empty sequence $\mathcal{R}_{\text{cand}}$ of candidate reward samples ;

**2 for** $(s,a) \in \mathcal{D}_{\text{eval}}$ **do**

**3** $\quad$ Sample a set of successor states $\mathcal{S}_{\text{succ}} = \{s'' \sim \hat{p}(\cdot|s,a)\}$;

**4** $\quad$ $R(s,a) = Q_{\theta_Q}(\phi(s), a) - \gamma \frac{1}{|\mathcal{S}_{\text{succ}}|} \sum_{s' \in \mathcal{S}_{\text{succ}}} \max_{a' \in \mathcal{A}} Q_{\theta_Q}(s', a')$;

**5** $\quad$ Append $R_t$ to $\mathcal{R}_{\text{cand}}$;

**6 end**

**7** Use samples to evaluate the prior $p_R(D_{\text{eval}}, \mathcal{R}_{\text{cand}})$ ;

**8** Use demonstrations to evaluate the likelihood $\mathcal{L}(\mathcal{D}|\theta_Q)$ per equation (5) ;

**Result:** unnormalized approximate posterior $p(\theta_Q|\mathcal{D}) \propto p_R(D_{\text{eval}}, \mathcal{R}_{\text{cand}}) p(\mathcal{D}|\theta_Q)$; candidate reward samples $\mathcal{R}_{\text{cand}}$.

---

match, we use it for the experiments in Section 4.3. In that case we require that $\mathcal{D}_{\text{eval}} \subseteq \mathcal{D}_+$, and for $s, a, s' \in \mathcal{D}_+$ we can define an empirical transition model $\hat{p}(s''|s,a) = \delta_{s'}(s'')$ to be used within the algorithm.

The corresponding continuous version of the algorithm is presented in Algorithm 2.

We can store both the Q function parameters $\theta_Q$ and the corresponding reward samples depending on downstream needs. We can then fit a warped Gaussian process to the posterior reward samples to get a posterior reward distribution over the whole state space. This can then be used together with an algorithm for RL (or *safe* RL in particular) to find an apprentice policy from the reward. Alternatively, as a shortcut, the posterior over Q-functions can be used to define an apprentice policy directly.

### 3.3 CONTINUOUS ACTIONS

The algorithm can be extended to continuous actions, replacing the sum in the Boltzmann likelihood (5 by an integral, and again, in turn, approximating it by a discrete set of samples from the action space. Simple discretizations (such as uniform sampling) can work well for low-dimensional action spaces (as we illustrate in our safe navigation experiment in the next section) but suffer from the curse of dimensionality, so a more sophisticated scheme would be

Table 1: **Speed comparison.** Samples per second produced by PolicyWalk and ValueWalk on a 3x3, 6x6, and 12x12 gridworld respectively.

| Num states | PolicyWalk | ValueWalk |
|---|---|---|
| 9 | 5.84 | 28.83 |
| 36 | 0.46 | 13.75 |
| 144 | 0.26 | 4.85 |

needed for higher-dimensional action spaces. We leave that for future work.

# 4 EXPERIMENTS

We tested our method on a small gridworld (for illustration and to compare the speed to PolicyWalk [Ramachandran and Amir, 2007], which our method builds upon but which is restricted to such small finite-space settings) and on 4 simulated control tasks with continuous states.

## 4.1 GRIDWORLD

For an illustration of the method with easily interpretable and visualizable features, we first test it on a simple gridworld environment shown in Figure 1. We have generated a fixed set of 50 demonstration steps in the environment and used our method, ValueWalk (including the environment dynamics), the original PolicyWalk [Ramachandran and Amir, 2007], and AVRIL [Chan and van der Schaar, 2021] (which does not use environment dynamics, making the comparison unfair but illustrative of inherent limitations of such model-free methods) to recover a posterior over rewards from an independent normal prior with mean 0 and standard deviation of 10. With the two MCMC methods, we took a total of 10,000 MCMC samples spread across 5 parallel chains using HMC+NUTS with 1000 warm-up steps per chain, which lead to $\hat{R} \leq 1.01$ on each dimension (where $\hat{R}$ is the potential scale reduction factor [Gelman and Rubin, 1992], a commonly used indicator that the chains have mixed well). We then also run the methods on a 6x6 and 12x12 version of the gridworld to examine how the compute times scale.

### 4.1.1 Results

Both PolicyWalk and ValueWalk (our algorithm) resulted in matching posterior reward samples as expected (confimed by two-sample Kolmogorov-Smirnov at $\alpha = 0.001$; comparison of their essentially same cdfs can be found in the supplement). The speed comparison of the two methods can be found in Table 1, showing ValueWalk indeed runs faster than the baseline PolicyWalk algorithm.

We also ran AVRIL on this simple grid world (which took 43s to converege). In terms of the resulting posterior, there are 3 things to note (see Figure 1 centre and right). Firstly, the posteriors are much tighter – the x-axis is zoomed in about 5x relative to the ValueWalk histograms. This is due to the fact that AVRIL does not model the uncertainty in the Q-function, instead learning only a point estimate. The reward posterior is then pegged to this Q-function point estimate thus significantly reducing its variance. As a result, both the reward of the obstacle and of the goal are extremely unlikely under the posterior.

Secondly, we can observe that the posterior reward for the obstacle is not any lower than that for most other states. This is because this state is never visited in demonstrations, and AVRIL – not taking the environment dynamics into account – consequently does not update this value. This illustrates an important downsides faced by methods without an environment model. (Note that the model-free version of ValueWalk would face the same issue.)

Finally, we can see that while the true posterior differs considerably from normal (see especially the strong skew of the negative-reward top middle cell), AVRIL is limited by its normal variational distribution. While in theory, AVRIL could be used with any variational family, we first need to determine which family may be suitable, for which an MCMC-based method such as ours is a useful instrument.

## 4.2 2D SAFE NAVIGATION ENVIRONMENT

To illustrate the potential of the full posterior over Q-values for synthesizing safe policies in a continuous-space environment, we also test our method on a simple 2D safe navigation environment. The state consists of a 2D position within the $[-10, 10]^2$ box, and the agent has a 2D continuous action at its disposal within an action space of $[-1, 1]^2$, which moves it by the given vector perturbed by white noise with standard deviation of 0.1. The region $[-9, 10]$ is a terminal goal state with a reward of 1; however, there is also a hazardous obstacle $[4, 6]^2$ with a penalty of -10. We collected 10 demonstration trajectories with $\alpha = 20$ and then ran ValueWalk and AVRIL on these demonstrations. We then used the two methods' estimates of the optimal Q-function to synthesize an apprentice policy. In AVRIL, we use the point estimate that the method learns. For ValueWalk, we tried policies optimizing the mean or the median of the Q-value estimates, but also a conservative policy maximizing the 0.1 quantile intended to have lower risk of low rewards.

### 4.2.1 Results

Figure 2 illustrates the demonstrations used and the learnt apprentice policies. Firstly, note that while the demonstrations are highly stochastic, the methods learn estimates of the *optimal* Q-function, thus possibly allowing them to pro-

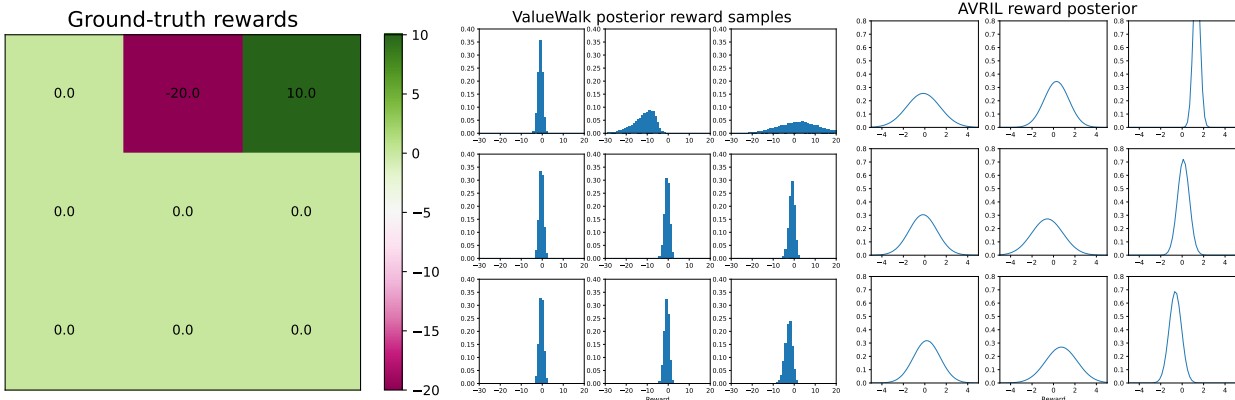

Figure 1: **Left**: Illustrative 3x3 gridworld. The agent always starts in the top left corner. The top right corner yields a reward of 10 and is terminal. The top centre tile represents an unsafe state that should be avoided and yields a reward of -20. **Centre**: Histograms of the samples from the posterior over rewards recovered by our ValueWalk corresponding to the 9 states of the gridworld. **Right**: Density functions of the posterior over rewards recovered by AVRIL.

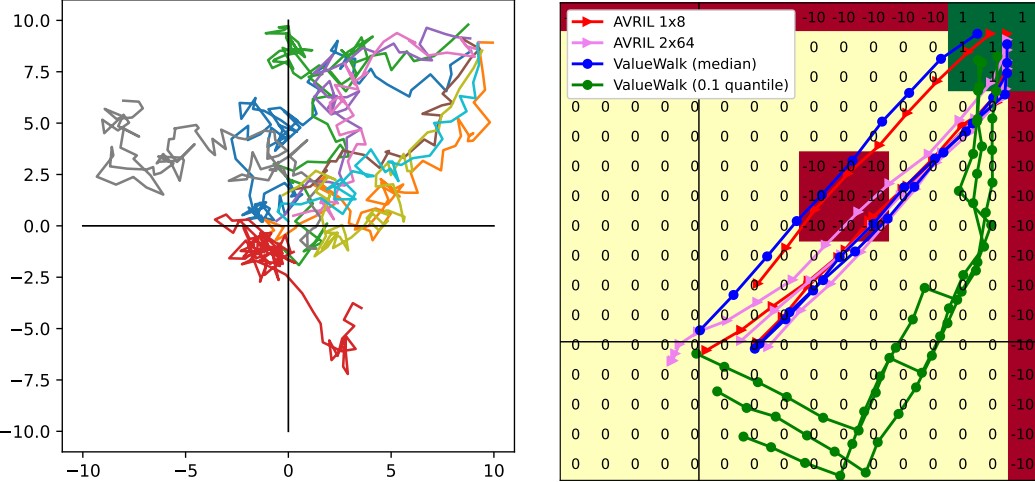

Figure 2: **Left**: The 10 demonstrations used in the continuous 2D environment. **Right**: Trajectories of policies derived from AVRIL and ValueWalk using an argmax of the inferred Q-values in each state. For AVRIL the Q-function point estimate is used. For ValueWalk, median and 0.1 quantile of the posterior distribution over optimal Q-values are used.

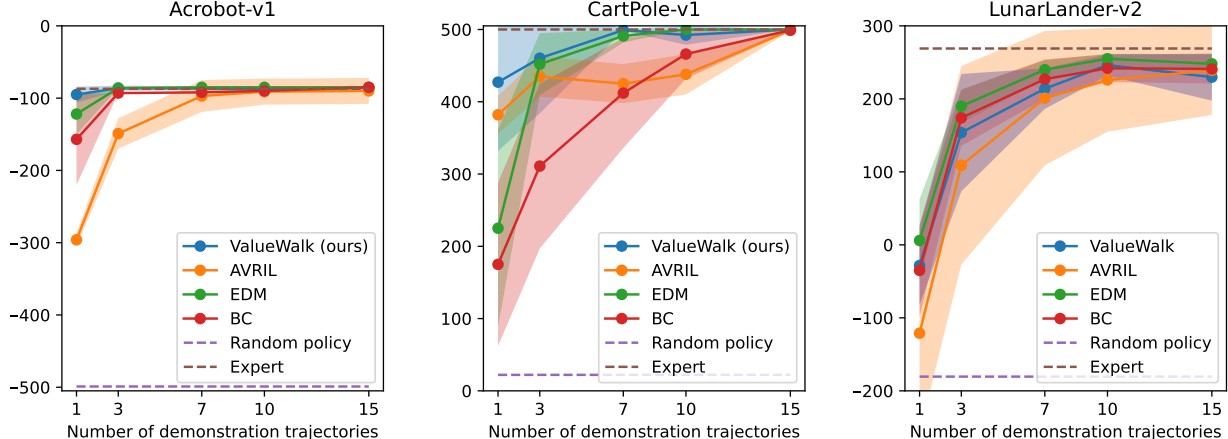

Figure 3: The test performance of an apprentice agent for ValueWalk and 3 baseline methods for different numbers of demonstration trajectories. The ValueWalk apprentice agent takes the action that maximizes the median of the posterior Q-value samples. The line shows mean performance across 10 runs with different sets of expert demonstrations; the shaded region shows mean±std.

duce apprentice policies with performance superior to that of the expert. Secondly, even to a human eye, the demonstrations leave it ambiguous whether there may be an unsafe region that the expert is avoiding, or whether the said area was missed by chance. While both the AVRIL apprentice policy, and the ValueWalk policies maximizing the mean and median of the Q-value distribution tend to go straight to the goal region (hitting the hazardous obstacle between 68 and 81% of cases), the 0.1-quantile-maximizing policy tends to avoid the region (hitting it in only 13% of cases across 100 sampled trajectories). This illustrates an important benefit of recovering a full posterior – it allows producing similar conservative policies based on statistics of the posterior distribution other than the usual mean.

## 4.3 CLASSIC CONTROL ENVIRONMENTS

To allow for direct comparison, we also evaluated Value-Walk on three classic control environments that were used to evaluate AVRIL by its authors: CartPole, where the goal is to balance an inverted pendulum by controling a cart underneath it, Acrobot, where the goal is to swing up a double pendulum using an actuated joint, and LunarLander, where the goal is to safely land a simulated lander on the surface of the moon. We used the same setup as was used for AVRIL to study the performance of an apprentice agent as a function of the number of demonstration trajectories for 1, 3, 7, 10, and 15 trajectories. The apprentice agent was evaluated on 300 test episodes and the mean reward is reported. We also compare against energy-based distribution matching (EDM; Jarrett et al. [2020]) – a successful method for strictly batch imitation learning – and plain behavioural cloning (BC) as a simple baseline. Baseline results were taken from Chan and van der Schaar [2021].

### 4.3.1 Results

The results are plotted in Figure 3. While both agents do close to expert-level when provided with 15 expert trajectories, our agent reaches this level with much fewer expert demonstrations. We hypothesize that this is due to treating the Q-function in a Bayesian way, as opposed to a point estimate in AVRIL, leveraging the advantages of a fully Bayesian treatment in the low data regime.

To support this, we can look at the log likelihoods of the action predictions on a hold-out set of 100 test trajectories and the entropies of the predictive posterior shown in Figure 4. For ValueWalk, the log likelihood increases as the method is given more trajectories, while the prediction entropy either decreases or stays about level as we would expect from a Bayesian method given increasing amounts of information. On the other hand, we do not consistently see similar behaviour in AVRIL. The test log likelihood consistently increases only in the case of the LunarLander environment, where it, however, starts from extremely low levels (the initial *mean* log probability of -18.0 would correspond to a probability of $10^{-8}$, suggesting the method has been putting practically 0 probability on actions taken by the expert among only 4 possible actions). Also, the prediction entropy of AVRIL tends to increase with seeing more trajectories. That suggests that AVRIL may be exhibiting overfitting behaviour in the low data regimes, which Bayesian methods should generally avoid.

The ValueWalk experiments were run until we get a well mixed chain, which can take between 4 and 38 hours of wall time on a single Nvidia RTX 3090 GPU[2] where AVRIL

---

[2]Experiments with fewer trajectories were run on a CPU.

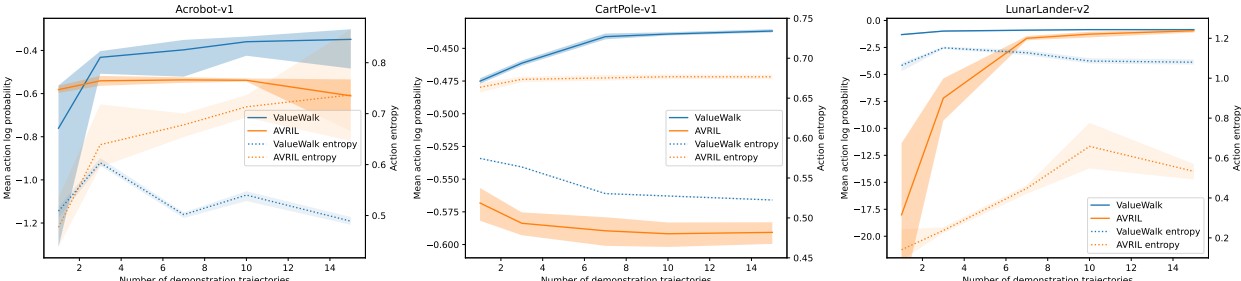

Figure 4: The log likelihood on a hold-out set of 100 test demonstrations and the entropy of the action predictions produced by ValueWalk and AVRIL.

takes 1-5 minutes to converge.

## 5 DISCUSSION

We presented a method that allows us to apply MCMC-based Bayesian inverse reinforcement learning to continuous environments. The method maintains the attractive properties of MCMC methods: it is agnostic to the shape of the posterior (where variational methods assume a particular parameterized distribution family) and given enough compute, produces samples from the true posterior. This comes at a large computational cost relative to cheaper methods, such as variational inference. However, we still think MCMC-methods do have a role to play in the Bayesian IRL ecosystem.

Firstly, we have shown that staying true to the Bayesian posterior does bring benefits in terms of superior performance on imitation learning tasks. Furthermore, the computational cost is paid in the learning phase, with inference at deployment being fast (sub millisecond per step in all cases, which would be sufficient for real-time control in most possible use cases and could be further optimized).

Secondly, we think that having a method that can draw samples from the true posterior can be extremely important in the process of developing other, faster or easier to scale methods, since it allows us to assess how their approximation deviates from the true posterior and how it impacts their performance. Also, variational methods in particular require a pre-specified family of distributions over which the optimization is subsequently run. ValueWalk can be used in an exploratory phase to determine what family of distributions may be appropriate for the problem at hand, before possibly using the advantages of variational methods to scale up.

Thus, despite their steep computational cost, we think MCMC methods have their place in Bayesian inverse reinforcement learning, and our method is a sizable step in extending them up to a wider range of settings.

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

# Walking the Values in Bayesian Inverse Reinforcement Learning (Supplementary Material)

**Ondrej Bajgar**[1]     **Alessandro Abate**[1]     **Konstantinos Gatsis**[2]     **Michael A. Osborne**[1]

[1]University of Oxford
[2]University of Southampton

## A   UNKNOWN TRANSITION PROBABILITIES

Section 3.1 presents a version of the ValueWalk algorithm for finite state and action spaces that assumes known transition probabilities. However, the key trick used in ValueWalk extends to unknown transition probabilities as well.

One simplified option to handle unknown transitions, also employed in the continuous-state case in Section 3.2 matching the setting used by AVRIL, is replacing the transition probabilities with their empirical estimate $\hat{p}(s'|s,a) = \xi(s,a,s')/\xi(s,a)$ where $\xi(s,a,s'), \xi(s,a)$ are the numbers of occurrences in the set of demonstration set of the transition $(s,a,s')$ and state-action pair $(s,a)$. In the finite-state, this would mean limiting the evaluation of the prior in Algorithm 3 to only those state-action pairs that do occur in the data (i.e. replacing vectors and matrices on lines 3-6 by the appropriate sub-vectors and sub-matrices).

A more principled Bayesian alternative is of course using full Bayesian inference also over transitions – in that case, we can perform the MCMC sampling jointly over both the transitions and the Q function parameters, recovering samples from the full joint posterior. The changes needed are (1) treating parameters of the transition model as inputs in the algorithm, (2) adding a prior over those parameters (so the joint prior will be a product of the Q-parameter prior and the transition-parameter prior), and (3) including transition probabilities in the likelihood. Here is the adaptation of the finite-space algorithm to this case of unknown probabilities:

---

**Algorithm 3:** Calculation of the unnormalized posterior for finite $\mathcal{S}$ and $\mathcal{A}$ with unknown transition probabilities (performed in each step of HMC). The resulting candidate reward sample $\bar{R}$ is then accepted/rejected together with the corresponding Q and P.

**Data:** a candidate matrix of Q values, a candidate transition matrix $P$, set of expert demonstrations $\mathcal{D}$, prior over rewards $p_R$, prior over transitions $p_P$

1 **for** $s, s' \in \mathcal{S}, a, a' \in \mathcal{A}$ **do**
2  $\quad$ $\pi(a|s) = \exp(\bar{\alpha}Q(s,a))/\sum_{a' \in \mathcal{A}} \exp(\bar{\alpha}Q(s,a'))$ ;
3  $\quad$ $\bar{P}(s,a;s',a') = P(s'|s,a)\pi(a'|s')$ ;
4 **end**
5 $\bar{R} = (I - \gamma\bar{P})\bar{Q}$ where $\bar{R}, \bar{Q}$ are flattened vector versions of the reward and Q-value matrices ;
6 $p_Q(Q) = p_R(\bar{R})\det(I - \gamma\bar{P})$ ;
7 $p(\mathcal{D}|Q) = \prod_{(s,a,s') \in \mathcal{D}} P(s'|s,a) \exp(\alpha Q(s,a))/\sum_{a' \in \mathcal{A}} \exp(\alpha Q(s,a'))$ ;
**Result:** $p(Q,P|\mathcal{D}) \propto p_P(P)p_Q(Q)p(\mathcal{D}|Q,P)$; candidate sample $\bar{R}$

---

*Accepted for the 40th Conference on Uncertainty in Artificial Intelligence* (UAI 2024).

# B PROOF OF SOUNDNESS OF THE ALGORITHM

**Theorem 1.** *Assume that the transition kernel $q_Q$ satisfies the detailed balance condition*

$$\frac{q_Q(Q'|Q)}{q_Q(Q|Q')} = \frac{p_Q(Q'|D)}{p_Q(Q|D)}$$

*with respect to the posterior over Q values defined in Algorithm 1. Then the associated implicit Markov chain over rewards also satisfies the detailed balance condition with respect to the posterior $p_R(R|D)$.*

*Proof.* Let $q_Q$ be the transition kernel over Q-values that satisfies the detailed balance condition with respect to the posterior $p_Q(Q|D)$ as assumed in the theorem statement.

The implicit transition kernel $q_R$ over rewards induced by $q_Q$ can be expressed as

$$q_R(R'|R) = q_Q(Q(R')|Q(R)) \left| \det\left( \frac{\partial Q(R')}{\partial R'} \right) \right| \tag{7}$$

where $Q(R) = (I - \gamma \bar{P})^{-1} R$ is the Q-value corresponding to reward $R$ as used in Algorithm 1. The determinant term accounts for the change of variables from $Q$ to $R$.

The posterior over rewards can be expressed in terms of the posterior over Q-values as

$$p_R(R|\mathcal{D}) = p_Q(Q(R)|\mathcal{D}) \left| \det\left( \frac{\partial Q(R)}{\partial R} \right) \right| = p_Q(Q(R)|\mathcal{D}) \left| \det(I - \gamma \bar{P})^{-1} \right|. \tag{8}$$

Now consider the ratio of the implicit transition kernel:

$$\frac{q_R(R'|R)}{q_R(R|R')} = \frac{q_Q(Q(R')|Q(R))}{q_Q(Q(R)|Q(R'))} \frac{\left| \det\left( \frac{\partial Q(R')}{\partial R'} \right) \right|}{\left| \det\left( \frac{\partial Q(R)}{\partial R} \right) \right|} = \frac{p_Q(Q(R')|D)}{p_Q(Q(R)|\mathcal{D})} \frac{\left| \det\left( \frac{\partial Q(R')}{\partial R'} \right) \right|}{\left| \det\left( \frac{\partial Q(R)}{\partial R} \right) \right|} =$$

$$\frac{p_R(R'|\mathcal{D}) \det((I - \gamma \bar{P}')^{-1})}{p_R(R|\mathcal{D}) \det((I - \gamma \bar{P})^{-1})} \frac{\det(I - \gamma \bar{P}')}{\det(I - \gamma \bar{P})} = \frac{p_R(R'|D)}{p_R(R|D)} \tag{9}$$

where the second equality follows from the assumed detailed balance condition on $q_Q$, the last equality follows from the expression for $p_R(R|D)$ derived above, and $\bar{P}'$ are the joint state-action transitions corresponding to $Q'$. Thus, the implicit Markov chain over rewards induced by the transition kernel $q_Q$ satisfies detailed balance with respect to the posterior $p_R(R|D)$, as claimed. $\quad\square$

The theorem establishes an important property of the ValueWalk method, namely that the implicit Markov chain over rewards induced by the HMC-based sampling of Q-values satisfies detailed balance with respect to the true posterior over rewards given the demonstrations, $p_R(R|D)$. This property is crucial for the soundness of the method.

Detailed balance is a sufficient condition for the Markov chain to have a stationary distribution equal to the target distribution, in this case $p_R(R|D)$. This means that, assuming the chain is ergodic, the samples of rewards obtained from the ValueWalk method will asymptotically follow the true posterior distribution, regardless of the initial distribution. In other words, the theorem guarantees that, given enough samples, ValueWalk will correctly characterize the posterior uncertainty over rewards, which is a key goal of Bayesian inverse reinforcement learning.

# C CONTINUOUS AVRIL

We are comparing ValueWalk with AVRIL Chan and van der Schaar [2021], which was originally designed to work with discrete actions. When we are comparing our method to AVRIL on continuous-action environments, we use the following continuous extension of AVRIL:

1. The original Boltzman likelihood 1 is replaced by its continuous version 5, which, in practice, gets calculated using the same approximation as our method.

2. Instead of taking the state as input and producing an output for each of the discrete actions, the Q function and the variational distribution for the reward now takes in a state-action pair (or a batch of those) and produces a single Q-value for those or a single set of variational distribution parameters.

# D  EXPERIMENT DETAILS

For the gridworld experiments, we used a version of AVRIL learning a Q-value for each state-action pair and a mean and variance value for the reward in each state. We use a matching setup for both ValueWalk and PolicyWalk.

In the continuous state space environments, for the 3 continuous baseline methods, we match the setup from Chan and van der Schaar [2021] and use neural network models with 2 hidden layers of 64 units and an ELU activation function. For our experiments, we scale up the network size with the complexity of the problem: we use one hidden layer with 8 units for the 2D safe navigation task and Cartpole, 1 layer of 16 units for Acrobot, and 2 layers of 32 units for LunarLander. In each case, we also tried running AVRIL with a matching network size but in each case it performed similarly or usually worse than the default 2x64 setup for which results are reported.

With ValueWalk, we use the Pyro [Bingham et al., 2018] implementation of HMC+NUTS, which we ran with 2,000 warm-up steps and 20,000 inference steps. We automatically tune the step size during warm-up but do not tune the mass matrix.

In the continuous environments, we use a Gaussian process prior with an RBF kernel with fixed scale of 1 and fixed lengthscale of 0.2 for Cartpole and Acrobot and 0.05 for Lunar Lander (chosen manually based on the distribution of features in each environment).

In Cartpole, Acrobot, and Lunar Lander, we reuse the demonstration sets provided by the authors of AVRIL. Each contains 1000 demonstration trajectories, from which we randomly chose a set of 100 test trajectories and then randomly sampled the reported numbers of train trajectories. We reran most experiments 10 times with different random sets of training trajectories and different random initializations.

Unless otherwise stated, we use a Boltzmann rationality coefficient of 1.

# E  ADDITIONAL DETAILS OF RESULTS

## E.1  GRIDWORLD EXPERIMENTS

Figure 5 shows the empirical cumulative distribution functions of the 10,000 posterior reward samples collected by PolicyWalk and ValueWalk and confirms both methods track the same posterior.

Figure 6 shows 2-D histograms of pairwise joint posteriors over rewards of the 9 states of the gridworld. Two aspects of the expert's behaviour are captured by this plot and may not be obvious from the simple histograms in Figure 1. Firstly, the agent heading to the terminal top right corner can be explained either by the reward there being positive, or by the reward in other states (especially the initial state) being negative and thus the agent using the terminal state as a way to escape incurring further negative rewards. Secondly, note that practically all of the probability mass is placed on the reward of the obstacle tile being lower than that of the two tiles below, thus explaining the expert avoiding the obstacle tile.

The plot also clearly shows that the posterior is non-Gaussian (note especially the sharp edge expressing high confidence that the ratio of the two values does not cross a certain threshold) and thus could not be captured by the Gaussian-assuming variational prior.

Note that this plot was produced with a prior standard deviation of 33 and an obstacle reward of -100.

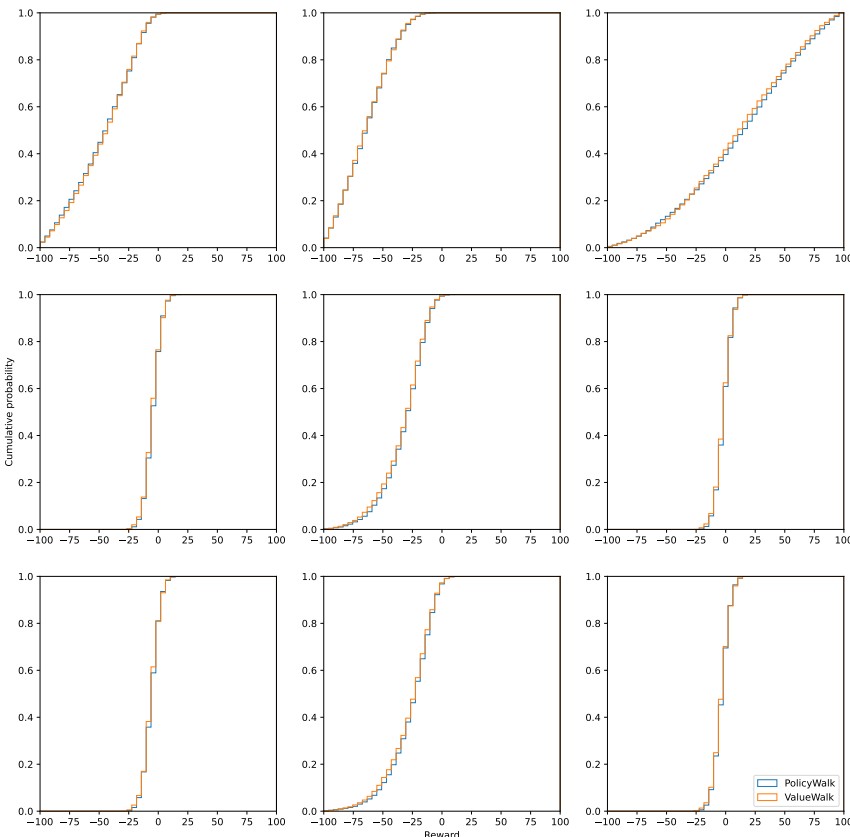

Figure 5: Cumulative distribution functions of the posterior distributions over rewards recovered by PolicyWalk and ValueWalk in the 3x3 gridworld, illustrating that the two methods recover the same posterior.

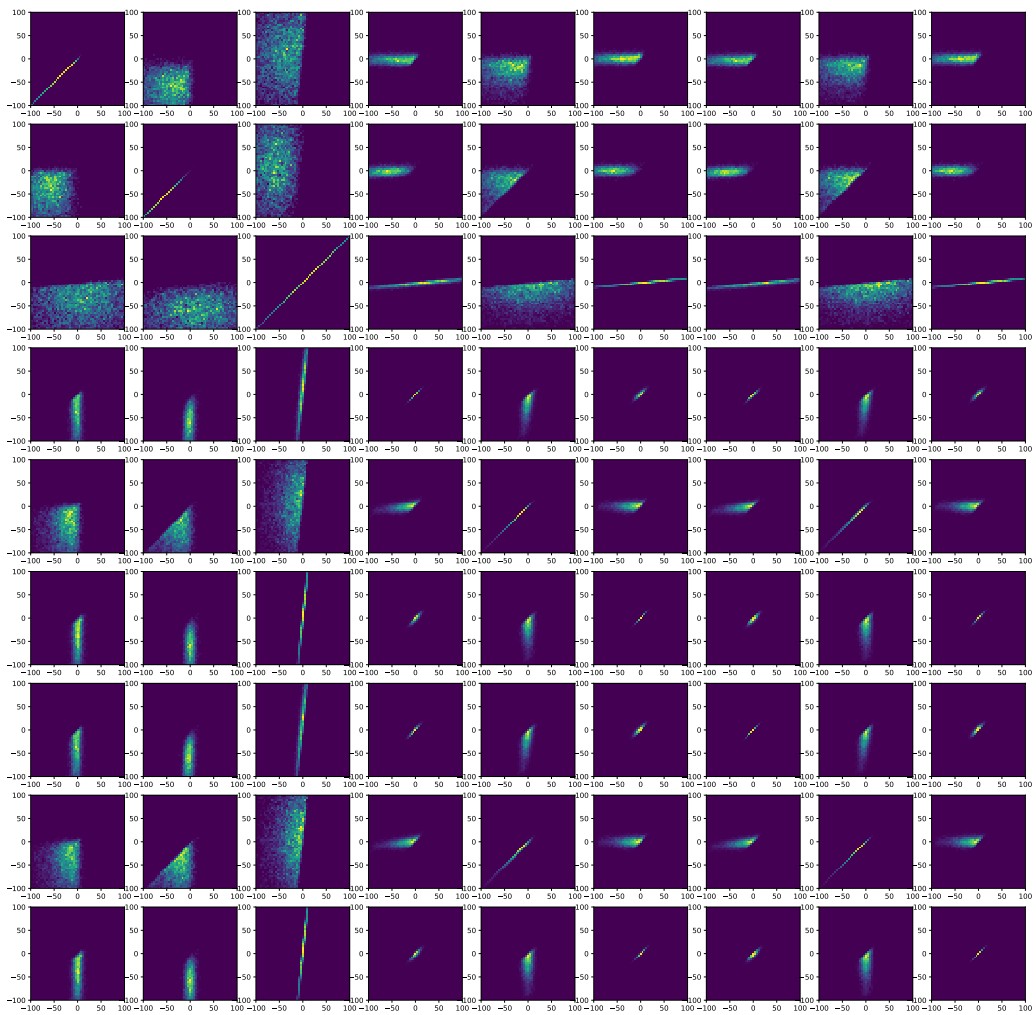

Figure 6: 2-D histograms representing the joint posteriors of the rewards associated with the 9 states of the gridworld (enumerated left-to-right, top-to-bottom, so state 3 is the goal state in the top right corner.

