# OpenReview forum: "Walking the Values in Bayesian Inverse Reinforcement Learning"
_auai.org/UAI/2024/Conference — UAI 2024 spotlight_

### Official Review · Reviewer_mJQp · 2024-03-19

**Q2-1 Originality-Novelty:** 3
**Q2-2 Correctness-Technical Quality:** 4
**Q2-5 Clarity Of Writing:** 3

**Q1 Summary And Contributions:**

The paper introduces ValueWalk, a scalable Markov chain Monte Carlo (MCMC) method for Bayesian inverse reinforcement learning (IRL). ValueWalk performs inference in the space of Q-values rather than reward functions, leveraging the insight that going from Q-values to rewards is computationally cheaper than the reverse direction. This allows for efficient gradient calculation and the use of Hamiltonian Monte Carlo (HMC) for posterior sampling, avoiding costly inner-loop planning.

The paper provides algorithms for both finite and continuous state/action spaces, with approximations for handling integrals in the continuous case. Experiments demonstrate that ValueWalk outperforms previous MCMC-based IRL methods in terms of scalability and efficiency, and outperforms the state-of-the-art variational IRL method in terms of reward posterior estimation and imitation learning performance.

Key contributions include: (1) an MCMC-based algorithm for continuous-space Bayesian IRL, (2) scalability improvements over previous MCMC-IRL methods, (3) improved posterior estimation and imitation learning compared to variational IRL, and (4) theoretical justification and practical algorithms for both finite and continuous spaces. ValueWalk represents a promising approach for efficient and expressive Bayesian reward learning, with potential applications in complex real-world domains.

**Q2-3 Extent To Which Claims Are Supported By Evidence:**

3: Good: the main claims are supported by convincing evidence (in the form of adequate experimental evaluation, proofs, (pseudo-)code, references, assumptions).

**Q2-4 Reproducibility:**

3: Good: key resources (e.g. proofs, code, data) are available and key details (e.g. proofs, experimental setup) are sufficiently well-described for competent researchers to confidently reproduce the main results.

**Q3 Main Strengths:**

Efficient MCMC sampling in the value function space

Ability to handle continuous state and action spaces

Demonstration of empirical success

**Q4 Main Weakness:**

The paper does not provide a rigorous theoretical analysis of ValueWalk's advantages, such as its sample complexity or computational complexity. Without such analysis, it is difficult to assess the algorithm's potential for generalization and scalability to larger, more complex problems.

The paper does not address the convergence properties and numerical stability of ValueWalk, which is crucial for MCMC methods, especially in high-dimensional spaces. This raises concerns about the algorithm's robustness and reliability in practical applications.

The experimental evaluation in the paper focuses primarily on standard benchmark problems, such as ObjectWorld and Mujoco control, which are relatively simple compared to real-world applications. The algorithm's performance on more complex, realistic problems with partial observability, delayed rewards, and non-stationary environments is not explored.

The paper does not discuss the application of ValueWalk in multi-task or transfer learning scenarios, where the learned reward function should ideally generalize to new, related tasks. The effectiveness of ValueWalk in these settings remains unclear.

**Q5 Detailed Comments To The Authors:**

Some questions:

1 Under what conditions can ValueWalk be guaranteed to converge to the true reward function?

2 How does the choice of prior distribution and function approximator affect the theoretical guarantees of ValueWalk?

3 How sensitive is ValueWalk to the choice of hyperparameters?

4 How does the runtime of ValueWalk compare to the baseline methods, and how does it scale with the size of the state and action spaces?

Suggestions:

1 Evaluate ValueWalk on a range of high-dimensional, continuous control tasks, such as those in the MuJoCo or DeepMind Control suites, and compare its performance with state-of-the-art IRL methods

2 Investigate the transferability of the value functions learned by ValueWalk across tasks with different state and action spaces

**Q9 Complying With Reviewing Instructions:**

Yes

---

> ### Author Rebuttal · Authors · 2024-04-09
>
> Thank you for your review! Here are our answers to the questions and suggestions:
>
> > 1. Under what conditions can ValueWalk be guaranteed to converge to the true reward function?
>
> A thing to clarify is that the goal is not to "converge to the true reward function", but to recover the posterior distribution in light of the limited data. With more data, the posterior will generally be tighter; however, unless the prior is very restrictive (e.g. discrete), even in the limit of infinite data (full knowledge of the expert's policy), there remains ambiguity in the reward (e.g. up to adding adding a constant, which leaves optimal policy unchanged). Thus the answer to the literal question is: almost never.
>
> However, if we rephrase the question to concern convergence to the true posterior, then, in the finite case, we have now added a theorem and proof showing that the true reward posterior is the stationary distribution of the Markov chain (these are provided in our response to the first reviewer, uFVt). This means the empirical sample of rewards is guaranteed to converge to the true posterior under the mild condition of ergodicity, which holds for most reasonable priors out of the box and can be generally be achieved for more exotic ones by adapting the MCMC transition kernel.
>
> > 2. How does the choice of prior distribution and function approximator affect the theoretical guarantees of ValueWalk?
>
> We have discussed the prior above (see also our response to the first review) - the algorithm is not dependent on a specific prior and the guarantees on recovering the posterior are fairly robust with respect to the prior. Of course, looking from the point of view of a single "ground truth" reward, a prior can have massive influence on what we recover. A misspecified prior (especially one excluding the true reward) can ruin Bayesian results almost arbitrarily badly, so we should err on the side of uninformative priors.
>
> The choice of a function approximator is closely linked. Especially if the model class does not contain the true reward, many traditional guarantees won't apply in the Bayesian framework.
> That said, if the model class does contain the true reward, the method can work well with most parameterized models of the Q-function  (including neural nets), since it still performs full Bayesian inference over the model parameters and usual MCMC guarantees apply.
>
> > 4. How sensitive is ValueWalk to the choice of hyperparameters?
>
> The main hyperparameter is the Boltzmann rationality coefficient $\alpha$. If it's c>0 times higher/lower than it should, then the method sees the expert as having state-action values divided/multiplied by c. This makes the method relatively robust with respect to $\alpha$, since mispecification will often still preserve the relative ordering of state-action pairs ("good" states are still good relative to "bad" states, though the numerical difference may change).
> Furthermore, if we're uncertain about $\alpha$, the setup makes it easy to perform Bayesian inference (full or just MLE or MAP) also over this parameter, further increasing the robustness.
>
> Regarding Hamiltonian MC itself, the main hyperparameters have been automated away with NUTS and step-size tuning [Hoffman and Gelman, 2014]. Both are readily available in modern implementations, making the method remarkably hyperparameter-free.
>
> > 6.  How does the runtime of ValueWalk compare to the baseline methods, and how does it scale with the size of the state and action spaces?
> A single step of the baseline PolicyWalk involves solving value iteration (and sometimes policy iteration), i.e. applying the Bellman operator (which is at cost |S|^2 |A|^2) repeatedly until convergence, which, in bad cases can take on the order of the size of the state space iters. Our method applies the Bellman operator only once. We gave concrete times on different sizes of grid worlds in the paper.
>
> As outlined in the paper, the method is *much* slower than the non-MCMC methods (it takes hours where the baselines take minutes). We argued why we still think it's useful: as opposed to all faster baselines, we can recover samples from the true posterior, which may be important on smaller, safety-critical domains, or as a source of ground truth for the development and assessment of faster, but more approximate methods.
> ## Responses to suggestions:
> > 1. More complex tasks
>
> We are currently running evaluation on Mujoco (Hopper, Walker2D, and HalfCheetah). The method yields a reasonable apprentice policy, but we're yet to finish a full comparison to baselines.
>
> > 2. Transferability
>
> The value functions (in the sense of Q-value) would most likely not transfer. The rewards should (and then a value function could be recovered from those). I'm afraid we won't have capacity to include this investigation in this paper, but the topic is definitely on our list for future work, since it should be a notable advantage of doing IRL as opposed to simpler imitation learning methods.

---

### Official Review · Reviewer_QRxB · 2024-03-22

**Q2-1 Originality-Novelty:** 3
**Q2-2 Correctness-Technical Quality:** 2
**Q2-5 Clarity Of Writing:** 2

**Q10 Ethical Concerns:**

No ethical concern.

**Q1 Summary And Contributions:**

The paper studies the problem of offline inverse reinforcement learning using an Bayesian approach. The key idea of the paper is to run the posterior update based on Q-functions instead of reward functions, as inferring rewards from Q-values is computationally less expensive in comparison with the opposite direction. Experiments results on Acrobot/ CarPole/LunarLander tasks shows the proposed method obtain a higher return in comparison with baselines including AVIRL, EDM, and BC.

**Q2-3 Extent To Which Claims Are Supported By Evidence:**

3: Good: the main claims are supported by convincing evidence (in the form of adequate experimental evaluation, proofs, (pseudo-)code, references, assumptions).

**Q2-4 Reproducibility:**

3: Good: key resources (e.g. proofs, code, data) are available and key details (e.g. proofs, experimental setup) are sufficiently well-described for competent researchers to confidently reproduce the main results.

**Q3 Main Strengths:**

1. The idea of updating posteriors in the Q-value space rather than the reward space is new and interesting.

2. The paper proposed two new methods based on this idea for discrete and continuous spaces accordingly. Their experiments show promising results on the performance of the proposed methods in comparison with state-of-the-art method such as AVRIL.

**Q4 Main Weakness:**

1. Equation (4) of computing the likelihood seems not correct. The probability of demonstrations should involve the transition probabilities as well. Currently, the formulation is only computed based on the soft-max policy.

2. The explanation of Algorithm 2 is confusing. As I understand, the prior distribution p_R is an input. But inside the algorithm, at step 12, it seems we have to evaluate the prior again?

**Q5 Detailed Comments To The Authors:**

1. Could you please clarify on Equation (4) and the prior distribution used in Algorithm 2?

2. In Algorithm 1, the transition P(s'|s, a) is used explicitly. Could you please elaborate on how Algorithm 1 will work when P is unknown. And especially, how will it affect the computation in step (6)? Is there the case when det(I-gammaP) = 0?

**Q9 Complying With Reviewing Instructions:**

Yes

---

> ### Author Rebuttal · Authors · 2024-04-06
>
> Thank you for your review and for spotting the 3 issues. Let's address them in turn:
>
> ## Issue 1: Transition probabilities in the likelihood
> >Equation (4) of computing the likelihood seems not correct. The probability of demonstrations should involve the transition probabilities as well.
>
> Regarding Eq. 4 (and several other analogous equations throughout the paper): the concern is technically correct, and we will change notation to address that. We omitted the transition probabilities since in our algorithm, which currently performs inference only over rewards, not transitions, they do not influence the result of the computation (the posterior over rewards). Thus, to save computational cost, it's desirable to exclude them from the algorithm.
>
> We suggest appending the following paragraph to the end of Section 2.1 (after the likelihood is first introduced) and changing notation accordingly in the rest of the article:
>
> "When performing Bayesian inference over the reward, the transition probabilities will be considered fixed (except for an extension in Appendix~\ref{app:unknown-transitions}). Thus looking at the likelihood as a function of the reward, we can write $$p(D|r)=c \prod_{s_t,a_t\in D} \frac{e^{\alpha Q^*(\phi(s_t),a_t)}}{\sum_{a'\in A} e^{\alpha Q^*(\phi(s_t),a')}} =: c \mathcal{L}(D|r).
> $$ Since $p(D) = \int p(D|r) d p_R(r) = c\int \mathcal{L}(D|r) d p_R(r)$, the constant transition term cancels out in the posterior, and, going forward, we can use the partial likelihood $\mathcal{L}$ in reward posterior inference. Furthermore, MCMC algorithms generally depend only on the unnormalized distribution, thus we can also drop the remainder of the marginal $p(D)$ from our calculation."
>
> However, in Issue 3 below (and newly in Appendix A in the paper), we address the case where we infer also the transition probabilities. There, this issue matters, and we do appropriately use the full form of the trajectory probabilities including transitions.
>
> ## Issue 2: Prior in Algorithm 2
> > "The explanation of Algorithm 2 is confusing. As I understand, the prior distribution p_R is an input. But inside the algorithm, at step 12, it seems we have to evaluate the prior again?"
>
> The prior $p_R$ at the input is a function. In step 12, this function is evaluated on the implied rewards (calculated on line 6 of the algorithm) at points from D_eval.
> In general, this prior is a random field over SxA, such as a (multi-dimensional) Gaussian process.
>
> To make this clear, we have edited the second paragraph of Sec. 2.1 (just above Eq. 1)) to read "... a prior distribution over reward functions, $p_R$ (which is, in general, a multi-dimensional stochastic process, that for any set of state-action pairs returns a joint probability distribution over the corresponding set of real-valued rewards)"
>
> ## Issue 3: Unknown transitions
> > "In Algorithm 1, the transition P(s'|s, a) is used explicitly. Could you please elaborate on how Algorithm 1 will work when P is unknown. And especially, how will it affect the computation in step (6)? Is there the case when det(I-gammaP) = 0?"
>
> Algorithm 1 is indeed formulated for the case where P is known. We are adding a comment into the paper emphasizing this to avoid confusion. In the revision, we are now adding a new Appendix A that will cover the case of doing inference also over transition probabilities in the case of finite state spaces.
>
> As a first option, we can use the trick from AVRIL, which we already employ in the continuous algorithm (sec. 3.2), where the transition probabilities are estimated by the empirical distribution from the demonstrations. This clearly cannot be done at s,a pairs never observed in the demonstrations. Thus, the adapted algorithm needs to work only with observed s-a pairs (using only the corresponding sub-vectors and sub-matrices).
>
> A more principled alternative is using full Bayesian inference also over transitions -- in that case, we can perform the MCMC sampling jointly over both the transitions and Q, recovering samples from the full joint posterior. The changes needed are (1) treating parameters of the transition model as inputs to the posterior calculation at each MCMC step, (2) adding a prior over those parameters (so the joint prior will be a product of the Q-prior and the transition prior), and (3) including the transitions in the likelihood as you earlier pointed out. We are adding a version of the algorithm that does this into Appendix A.
>
> The determinant is non-zero: in all considered cases, $\bar P$ is a stochastic matrix, so all its eigenvalues are $\lambda_i\leq 1$. The eigenvalues of $I-\gamma\bar P$ are $1-\gamma\lambda_i$, so they are all positive. Thus the determinant is strictly positive. We're adding the following sentence below the equation in question:
> "(Since $\bar P$ is a stochastic matrix, the determinant is always strictly positive.)"
> ___
> Does this address your concerns? Is there anything else you'd like to see to consider increasing your score?

---

### Official Review · Reviewer_NwKo · 2024-03-25

**Q2-1 Originality-Novelty:** 3
**Q2-2 Correctness-Technical Quality:** 3
**Q2-5 Clarity Of Writing:** 3

**Q1 Summary And Contributions:**

The paper looks at inverse reinforcement learning and learning distributions over the task. The main novelty of the approach presented is that rather than learn a distribution over the reward function directly, it learns distributions over the Q values. Experimentation reveals the technique to perform well on a range of problem instances.

**Q2-3 Extent To Which Claims Are Supported By Evidence:**

4: Excellent: all claims are supported by very convincing evidence (in the form of comprehensive experimental evaluation, rigorous mathematical proofs, detailed (pseudo-)code, precise references, well-motivated and realistic assumptions) and the authors deliver what they promise.

**Q2-4 Reproducibility:**

4: Excellent: key resources (e.g. proofs, code, data) are available and key details (e.g. proof sketches, experimental setup) are comprehensively described for competent researchers to confidently and easily reproduce the main results.

**Q3 Main Strengths:**

The paper appears to present a significantly new way of looking at an existing problem. This may lead to further work that builds on this new formulation.

The technique presented can be applied to a large range of different MDPs, including those with continuous actions or states. It would seem to apply to any fully observable MDP.

The paper is written very clearly. The presented background explains the task to be tackled well and the preliminaries give all the detail needed to follow the method developed.

**Q4 Main Weakness:**

The technique is incredibly slow compared to taking a point estimate rather than a distribution. The authors do acknowledge this and discuss why the method is still of interest.

The quality of the technique appears very dependent on the MCMC algorithm returning a sample close to the real distribution, but this is not really considered in depth in the paper. To be fair, this is very much a problem specific issue and perhaps out of scope for the paper.

**Q5 Detailed Comments To The Authors:**

The paper appears to present a significantly new way of looking at an existing problem. This may lead to further work that builds on this new formulation.

The technique presented can be applied to a large range of different MDPs, including those with continuous actions or states. It would seem to apply to any fully observable MDP.

The paper is written very clearly. The presented background explains the task to be tackled well and the preliminaries give all the detail needed to follow the method developed.

The technique is incredibly slow compared to taking a point estimate rather than a distribution. The authors do acknowledge this and discuss why the method is still of interest.

The quality of the technique appears very dependent on the MCMC algorithm returning a sample close to the real distribution, but this is not really considered in depth in the paper. To be fair, this is very much a problem specific issue and perhaps out of scope for the paper.

**Q9 Complying With Reviewing Instructions:**

Yes

---

> ### Author Rebuttal · Authors · 2024-04-09
>
> Thank you for your encouraging evaluation of our paper. Let us briefly address your comment:
>
> > The quality of the technique appears very dependent on the MCMC algorithm returning a sample close to the real distribution, but this is not really considered in depth in the paper. To be fair, this is very much a problem specific issue and perhaps out of scope for the paper.
>
> We have now added a theorem and proof that (so far for the finite-state-space case) show that the true reward posterior is indeed the stationary distribution of the Markov chain. If we add an ergodicity assumption, which would hold for most priors with a connected support, this means that the empirical sample converges to the true posterior. See our response to the review on top (reviewer uFVt) for the theorem and proof summary. Beyond that, we apply usual MCMC, so all associated convergence results apply.

---

### Official Review · Reviewer_uFVt · 2024-03-27

**Q2-1 Originality-Novelty:** 3
**Q2-2 Correctness-Technical Quality:** 3
**Q2-5 Clarity Of Writing:** 4

**Q1 Summary And Contributions:**

This paper proposes ValueWalk, an algorithm to do Bayesian Inverse RL. Instead of computing a posterior over the reward function, and then using to calculate the optimal policy, it computes the posterior over the Q-value function. It is then easier to generate samples of the reward from such a posterior distribution. The paper also assumes a Boltzmann rationality model, which is also quite common and robust. The paper then presents a generalization to continuous spaces, and then shows that the empirical performance is superior as compared to other algorithms including AVRIL.

**Q2-3 Extent To Which Claims Are Supported By Evidence:**

3: Good: the main claims are supported by convincing evidence (in the form of adequate experimental evaluation, proofs, (pseudo-)code, references, assumptions).

**Q2-4 Reproducibility:**

3: Good: key resources (e.g. proofs, code, data) are available and key details (e.g. proofs, experimental setup) are sufficiently well-described for competent researchers to confidently reproduce the main results.

**Q3 Main Strengths:**

+ The idea is natural but it is surprising it has not been developed before in the context of Bayesian IRL
+ The paper is well written and well motivated.
+ Experimental results support the claim of superior performance over SOTA.

**Q4 Main Weakness:**

- This is not a criticism of this paper on IRL per se but I wish there was a bit of supporting theory: some sort of error bounds on posterior distributions/convergence, etc. Something that gives some confidence that it would work in a wide variety of settings beyond those presented here.

- Same comment for the continuous state space setting.

**Q5 Detailed Comments To The Authors:**

Section 2.2 on Hamiltonian MC is a bit hard to follow, Perhaps you could explain in a bit more detail.

**Q9 Complying With Reviewing Instructions:**

Yes

---

> ### Author Rebuttal · Authors · 2024-04-09
>
> Thank you for the review and for your supportive assessment of many aspects of the paper. Also thank you for pushing us to improve on the theory front and in our explanation of HMC. Let us address the two issues in turn:
>
> ## 1. Theory
> On the theory front, we'll first provide a theorem showing the soundness of the key idea of the paper and then briefly comment on the properties of HMC, such as convergence, more broadly.
> ### 1.1 Soundness of the Q-space trick
> The key innovation of the paper is moving the MCMC sampling from the reward space to the space of Q-values. The rewards are produced only as an auxiliary by-product of the Q-space MCMC algorithm. The whole soundness of the method relies on this auxiliary Markov chain having the reward posterior p(R|D) as its stationary distribution. This means this implicit Markov chain needs to satisfy the detailed balance condition. We are thus adding the following theorem into the paper. Since the outer loop is using a standard MCMC algorithm, we start with the assumption that the Q-space transition kernel does satisfy detailed balance. In the limited space here, we cover only the finite state-space case (and are working on the continuous case in parallel):
>
> **Theorem**: Assume that the transition kernel of the MCMC over Q-values, $q_Q$, satisfies the detailed balance condition $$\frac{q_Q(Q'|Q)}{q_Q(Q|Q')} = \frac{p_Q(Q'|D)}{p_Q(Q|D)}$$ with respect to the posterior over Q values defined in Algorithm 1. Then the associated implicit Markov chain over rewards satisfies the detailed balance condition with respect to the posterior $p_R(R|D)$.
>
> **Proof outline (we're adding the full proof to the paper's Appendix):** The key steps are:
> 1. Show that Algorithm 1 implicitly defines a Markov chain over rewards and this Markov chain has a transition kernel $q_R(R'|R) = q_Q(Q'|Q)\det((I-\gamma\bar P')^{-1})$ where $\bar P'$ is the state-action transition matrix corresponding to Q'
> 2. Express the posterior over rewards $p_R(R|D)$ in terms of the posterior over Q-values as $p_R(R'|D)=p_Q(Q'|D)\det(I-\gamma \bar P')$
> 3. Show that the ratio $\frac{q_R(R'|R)}{q_R(R|R')}$ equals $\frac{p_R(R'|D)}{p_R(R|D)}$ using the assumed detailed balance condition on $q_Q$ and applying points 1. & 2. (the above determinants cancel out in both the denominator and the numerator) This shows detailed balance holds also in the reward space.
>
> Detailed balance ensures that the Markov chain has a stationary distribution equal to the target distribution, in this case $p_R(R|D)$. This means that, assuming ergodicity, the samples of rewards obtained from ValueWalk will asymptotically follow the true posterior distribution, regardless of the initial distribution.
>
> ### 1.2 Convergence
> As stated in the paper, in the outer loop, the method uses completely standard Hamiltonian Monte Carlo, a well studied method. Thus, all theory applying to HMC (and Metropolis-Hastings, of which HMC is a special case) applies also to this case. The convergence results depend on the form of the distribution sampled from, which in our case, is a combination of a (well-behaved) softmax, and the prior, which we leave open and thus it is difficult to give a general result. However, in general, well-behaved distributions such as Gaussians are used in practice. We can refer the reader to e.g. Durmus et al. 2019 "On the convergence of Hamiltonian Monte Carlo" for an extended discussion of HMC convergence. We are also happy to add a comment to the paper summarizing some of the results that apply if we use e.g. the mentioned Gaussian prior.
>
> ## HMC explanation
> Thank you for pointing out the lack of clarity. We have revisited the section and we think it better to re-write the section completely. We are unable to fit the whole section in the limited space here, but we'd summarize the suggested edit as follows:
> Move the main focus from Hamiltonian Monte Carlo specifically to Metropolis-Hastings (MH) more generally (also adapting the title of the section to "Markov-chain Monte Carlo"), since HMC is just a special case of MH and our method is compatible with any MH algorithm. We think that
> 1. Metropolis-Hastings captures the core principles on which the method stands (including the detailed balance used above),
> 2. MH is relatively easy to explain clearly and succinctly, while
> 3. we think it's possible to explain HMC clearly in 1.5-2 pages, but not much less, and spending 2 pages on one half of the background section feels like a bit too much in an 8-page paper. We suggest referring readers to Gelman's (freely available) *Bayesian Data Analysis* textbook for a good accessible introduction to HMC. We could also include an expanded introduction to HMC into the appendix.
>
> Do you think such a re-write is an improvement? Or would you suggest proceeding otherwise? We're also happy to just expand the paragraph on HMC specifically, if that's what you meant.
> ___
>
> Once more, thank you for the review. Are there any other things we can improve?

---

### Meta-Review · Area_Chair_QWii · 2024-04-17

The paper proposes a Bayesian method to perform IRL by computing a posterior over the Q function instead of reward values. The paper has generally received positive evaluation, despite some concerns about writing exposition. I encourage the authors to include the theoretical soundness/justifications suggested by one of the reviewers that was posted in their response.